# The impact of COVID-19 on trips to urban amenities: Examining travel behavior changes in Somerville, MA

**Andres Sevtsuk**[1]*, **Annie Hudson**[2], **Dylan Halpern**[3], **Rounaq Basu**[1], **Kloe Ng**[1], **Jorrit de Jong**[4]

1 Department of Urban Studies and Planning, Massachusetts Institute of Technology, Cambridge, Massachusetts, United States of America, 2 Mobility Initiative, Massachusetts Institute of Technology, Cambridge, Massachusetts, United States of America, 3 Center for Spatial Data Science, University of Chicago, Chicago, Illinois, United States of America, 4 Kennedy School of Government, Harvard University, Cambridge, Massachusetts, United States of America

* asevtsuk@mit.edu

## Abstract

While there has been much speculation on how the pandemic has affected work location patterns and home location choices, there is sparse evidence regarding the impacts that COVID-19 has had on amenity visits in American cities, which typically constitute over half of all urban trips. Using aggregate app-based GPS positioning data from smartphone users, this study traces the changes in amenity visits in Somerville, MA from January 2019 to December 2020, describing how visits to particular types of amenities have changed as a result of business closures during the public health emergency. Has the pandemic funda-mentally shifted amenity-oriented travel behavior or is consumer behavior returning to pre-pandemic trends? To address this question, we calibrate discrete choice models that are suited to Census block-group level analysis for each of the 24 months in a two-year period, and use them to analyze how visitors' behavioral responses to various attributes of amenity clusters have shifted during different phases of the pandemic. Our findings suggest that in the first few months of the pandemic, amenity-visiting preferences significantly diverged from expected patterns. Even though overall trip volumes remained far below normal levels throughout the remainder of the year, preferences towards specific cluster attributes mostly returned to expected levels by September 2020. We also construct two scenarios to explore the implications of another shutdown and a full reopening, based on November 2020 con-sumer behavior. While government restrictions have played an important role in reducing visits to amenity clusters, our results imply that cautionary consumer behavior has played an important role as well, suggesting a likely long and slow path to economic recovery. By drawing on mobile phone location data and behavioral modeling, this paper offers timely insights to help decision-makers understand how this unprecedented health emergency is affecting amenity-related trips and where the greatest needs for intervention and support may exist.

**Data Availability Statement:** All relevant data are within the manuscript and its Supporting Information, and have also been uploaded to http://

cityform.mit.edu/projects/impact-of-covid-19-on-trips-to-urban-amenities.

**Funding:** The research was funded by the Bloomberg Harvard City Leadership Initiative (project name: Accessing urban amenities: implications of alternative business re-opening strategies in post-COVID19 cities, grant nr: 031415-0001), which receives financing from Bloomberg Philanthropies. Bloomberg Philanthropies played no role in the study design, data collection and analysis, decision to publish, or preparation of the manuscript.

**Competing interests:** The authors have declared that no competing interests exist.

## 1. Introduction

Since the COVID-19 pandemic spread across the US in early 2020, there has been much speculation about how it has reshaped both residential and work locations [1, 2]. Yet more than half of all trips made in American cities are not work trips, but rather journeys for shopping, personal errands, or for social and family purposes [3]. Amenity clusters—agglomerations of retail, food and beverage, personal service, and entertainment establishments—where many of these trips take place, play an important role in shaping daily urban mobility patterns [4]. While people cannot easily pick an alternative to a closed school or workplace, the landscape of urban amenities offers wide-ranging choices, visits to which can reveal how people obtained necessary goods and services during the pandemic, and how these choices compare to more normal times [5].

E-commerce had already substantially affected amenity visits before the pandemic. In the second quarter of 2019, a year before the pandemic, online retail sales constituted around 11% of retail sales. In the same quarter in 2020, the e-commerce market share rose to 16% [6]. But in addition to accelerating e-commerce orders that were already rising before the pandemic, COVID-19 may have not only reduced brick and mortar store visits across the board, but also reshaped consumer preferences as to which kinds of stores to patronize, how frequently, and at what locations.

In this study, we investigate how the public health emergency has reshaped preferences for local amenity-oriented trips, affecting how many trips, how far, and what types of amenity clusters are visited. We use anonymous mobile-phone global positioning system (GPS) data released by SafeGraph—a leading mobile phone data analytics company—to examine how visits to amenity clusters changed during different months of 2020—during the COVID-19 pandemic—as compared to the same months the year before in 2019. Despite the shortcoming of only aggregated data availability in anonymized records of mobile phone movements, big data from SafeGraph offers the potential for unorthodox, yet large-scale insights into urban mobility dynamics.

Changes to amenity-visit counts can result from three potential sources: changes in the built environment, changes in business regulations, and changes in preference (or behavior) among store visitors. Built environment changes originate from a natural evolution of cities and amenity clusters—spatial shifts in housing or employment patterns (trip origins) or shifts in store locations (trip destinations). Around 10% of a city's retail and service amenities naturally turn over in a typical year [7]. Second, while the regulatory climate that governs stores is relatively stable in a typical year (absent significant tax changes), COVID-19 has brought about major shifts in business regulations. A large number of businesses were forced to close when the State of Massachusetts ordered a shutdown for non-essential businesses and prohibited gatherings of more than ten people on March 24th, 2020 (See policy timeline in Supp. Materials A in S1 File, and [8, 9]). Business closures have significantly reduced the number of potential destinations patrons can visit during the pandemic.

Third, changed consumer preferences could result in patronage changes among businesses that did remain open, including which locations were patronized and how frequently during the shutdown order and the months that followed. It is specifically this shift in patrons' preferences that we aim to examine. Visitors might ordinarily prefer larger stores or more diverse clusters with more goods to reduce overall trips. Has the COVID-19 pandemic altered visitor preferences with respect to destination attributes? If so, has the behavioral shift changed during different phases of the pandemic? And finally, what can we say about longer-term effects of the pandemic on amenity-oriented trips—have the behavioral shifts remained consistent over time, or are visitors' preferences returning back towards pre-pandemic levels?

We use a behavioral modeling framework originally developed for aggregate destination choice analysis [10–13], to examine how specific attributes of urban retail clusters and travel distances have affected patronage probabilities of amenity clusters before and during the pandemic. The directionality and magnitude of these effects is examined at the Census block group (CBG) level. Having calibrated preference coefficients before and during the pandemic, these models also allow us to explore the effects of two potential scenarios under new behavioral habits: a) how a new closure of businesses in November 2020 could have impacted cluster visits differently than the first round of store closures that took place in April 2020, and b) what new, post-COVID-19 visiting patterns might look like with a full reopening of businesses.

Efforts to understand retail patronage behavior date back to the 1930s [14], and cover a wide range of elements, including product price [15, 16], store convenience [11, 17], service quality [18, 19], and demographic characteristics [20, 21], among others [22–25]. Literature exploring how retail or amenity visits may be affected by a pandemic or crisis is expectedly sparser. Some analysis examines how both long-distance [26–28] and local [29] travel behaviors might be affected by health outbreaks. McKercher [27], for example, argues that hysteria induced by SARS resulted in irrational travel behaviors and amenity patronage. Fenichel et al. [26], meanwhile, found an increase in missed flights during the H1N1 pandemic, pointing towards behavioral travel changes. At the local level, Kim et al. [29] used data from Seoul's smart cards to show that both destination characteristics and the presence of a MERS outbreak affected transit ridership and destination choice.

There is an emerging body of work on travel behavior changes sparked by COVID-19 to which this paper also contributes [5, 18, 30–36]. Shamshiripour et al. [5], for example, explored changes in behavior based on a stated-preference survey, anticipating large-scale behavioral changes as habits and priorities change during the pandemic. Pantano et al. [35] offer an overview of the changing retail landscape resulting from COVID-19, proposing potential interventions and policies to help ensure retailers' survival over the short and long terms. Basu and Ferreira [30] show that zero-car households are contemplating purchasing private cars due to fear of shared modes such as mass transit and ride-hailing, thereby implying significant post-pandemic shifts in travel behavior.

There is a gap in the literature regarding understanding the nuances of patronage as affected by a health outbreak on amenity visits in particular, which constitute roughly a half of all trips combined, based on national travel trends for shopping, social and errand destinations [3, 18, 35]. While changes to travel behavior have been explored alongside changes in consumer behavior, the two have rarely been linked to offer a more comprehensive view of how patronage of local main streets and amenity clusters has changed during COVID-19 and what ripple effects could be anticipated looking towards the future. This paper aims to address this gap, examining how COVID-19-induced behavior changes have affected visits to amenity clusters and how such behavior changes have shifted over successive months during the pandemic. The findings are relevant to municipal economic development departments as well as business associations and business improvement districts that coordinate policy and investments at the business cluster level [34, 37, 38].

The paper is structured as follows. The research methods section describes the data, the study area and our analysis framework. The results section describes how specific cluster attributes have impacted visitors' behavior before and during the pandemic and illustrates scenarios, which examine how the findings can be used to forecast potential policy impacts on amenity trips. The discussion outlines takeaways, shortcomings and questions for future work.

## 2. Research methods

### 2.1. Data

We use mobile phone location data for our analysis in order to better understand movement and patronage on a shorter timeline. Data regarding destination visits are taken from the Safe-Graph COVID-19 Data Consortium and contain information such as aggregated visitor counts to individual amenities from Census block groups as well as dwell time at each destination. SafeGraph receives raw GPS data from multiple different mobile app providers on both iOS and Android systems. We supplement these with ESRI Business Analyst data, which detail the store types and sizes present in each amenity cluster, and Somerville City data on outdoor seating and parking.

Since SafeGraph estimates visits to amenities based on smartphone GPS movements, the data does not cover all actual visitors but rather a subset of users that have a smartphone and who have enabled their phone's GPS location feature in various apps. The data could thus be under-representing visitors from demographic groups that have a lower proclivity to own or use a smartphone (e.g., elderly individuals and low-income residents). Yet, penetration is likely to be higher in urban and suburban areas, especially those with higher incomes, younger populations and higher education levels than the national average, such as Somerville MA (Supp. Materials B in S1 File).

Although SafeGraph provides estimates of visitors at specific establishments (e.g., a particular Starbucks), there is considerable potential for location error when such positioning is performed in dense urban amenity clusters. When stores are side by side, app-based GPS positioning is not able to distinguish with a high degree of certainty whether a user is in one store or another; under ideal circumstances, mobile GPS positioning can only achieve an accuracy of roughly 5 meters [39]. We therefore implemented our analysis not at the individual establishment level, but at an establishment cluster level.

Clusters, shown in Fig 1, were defined as groups of businesses that contain at least ten establishments and wherein each establishment is no more than 100 meters from its nearest neighboring establishment along the street network. This clustering approach has been previously used by Sevtsuk [4] and shown to identify agglomerations of amenities around intuitive "places" (e.g., Harvard Square, Porter Square etc.). We consider "amenities" to include North American Industry Categorization System (NAICS) codes 44–45 (retail), 722 (food and beverage), 811–812 (personal services), and 491, 7111, 712 or 713 (entertainment).

### 2.2 Study area

Our study area includes the City of Somerville and its immediate surroundings using a two-kilometer buffer. We obtained 1,457 unique destinations from the SafeGraph dataset, out of which 1,147 (78%) are retail, personal service, food and beverage or entertainment type amenities, commonly found along Somerville's streets and "squares".

Fig 2 illustrates the distribution of trips from block groups to amenity clusters in ten time periods: February, April, July, September and December 2019, and the same months in 2020. The 2020 observations, except February, fall under the COVID-19 pandemic. A visual comparison of trip distribution in February 2019 and 2020 (a month before the pandemic) suggests a broadly similar pattern in both years. As expected, during "normal" times, the amenity-oriented trip distribution does not vary significantly from one year to another. One notable difference in February 2020 is an increase in trips to Assembly Square—a newly developed retail center that expanded the number of stores and grew in popularity over the course of the year.

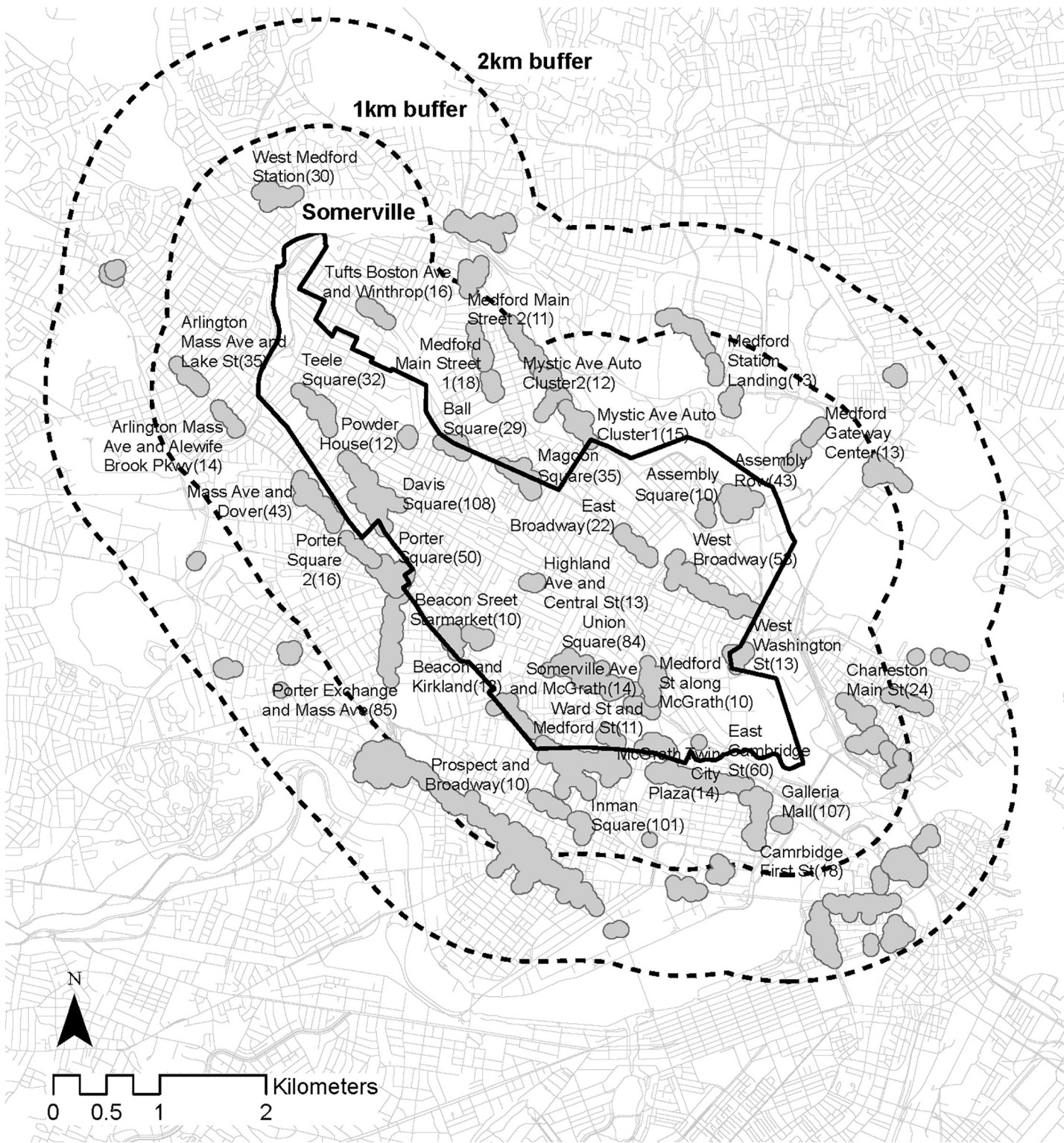

**Fig 1. Somerville, MA study area, showing 1km buffer containing CBGs and a 2km buffer containing amenity clusters.** Base map and data from OpenStreetMap and OpenStreetMap Foundation.

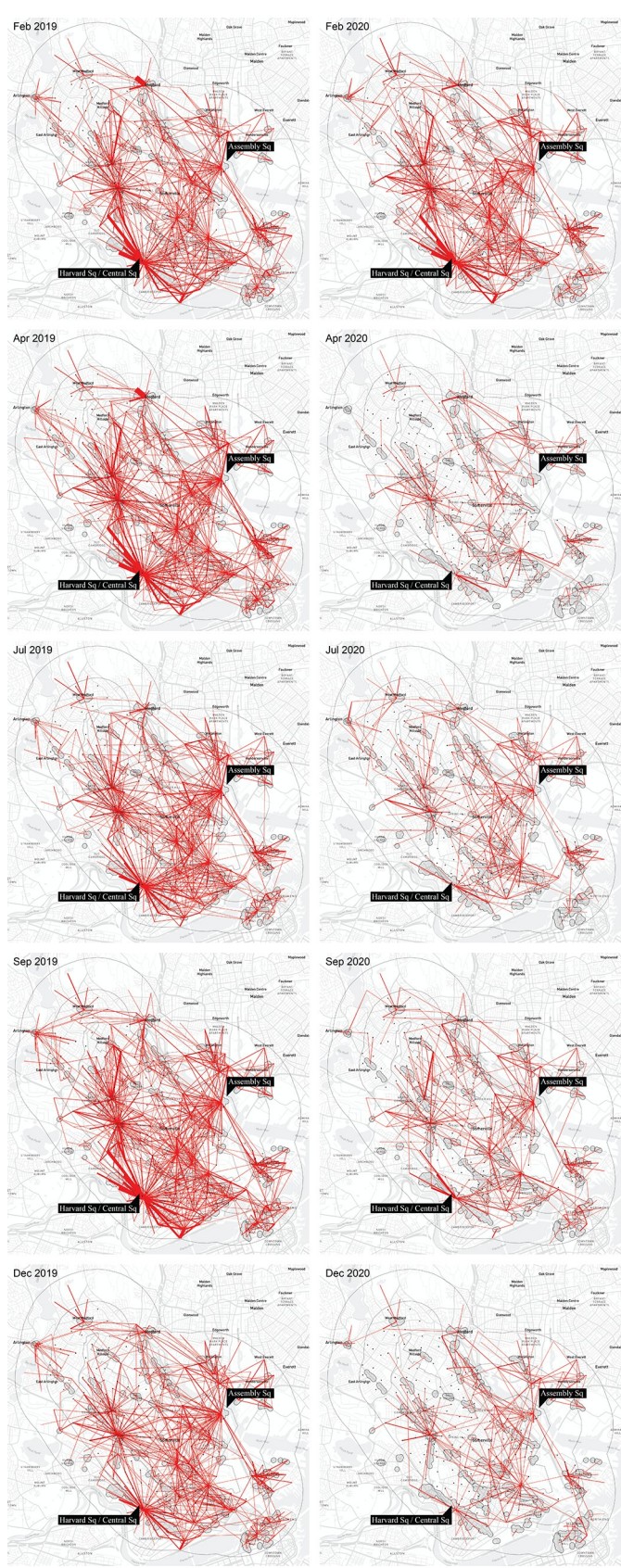

**Fig 2. Distribution of trips from CBGs in Somerville and a 1km buffer around it to amenity clusters in Somerville and a 2km buffer around it in February, April, July, September and December in 2019 (left) and 2020 (right).** Trips up to 3km in length. Base map and data from OpenStreetMap and OpenStreetMap Foundation.

The starkest year-on-year contrast in Fig 2 is captured between April 2019 and April 2020, when we find an 86% decrease in visits across all business establishments reported by Safe-Graph. Within Somerville, SafeGraph data reports a -65% mean decrease in activity comparing March through May 2019 to the same months in 2020 across the 12 most frequently visited NAICS 3-digit business categories. The most heavily impacted individual business types included furniture stores (NAICS 442), clothing stores (NAICS 448), and hotels (NAICS 721), which lost 71–78% of their visits compared to the same period the year before. Hardware and home improvement stores (NAICS 444) and grocery stores (NAICS 445) were some of the least impacted types of establishments losing 29% and 46% of visits, on average, respectively.

Fig 3 shows the mean change in visits to clusters for each month between 2019 and 2020. Each gray dot represents a particular cluster, with its position above or below the zero percent baseline marking the magnitude of change in the same month between 2019 and 2020. Though overall visits reached a bottom in April 2020 at -86%, visits remained low even in December 2020, at around -70% as compared to the year before. The vast majority of clusters saw large decreases, but the chart also suggests there were a handful of clusters that actually performed better in some months of 2020. These tend to be clusters in dense residential areas that benefited from a home-based population.

Fig 3 also maps the mean percent change in visits between 2019 and 2020 observed across all pandemic months (April-December) spatially. The clusters with the largest observed decreases in visits are near job centers and university campuses in Cambridge—Harvard, Central and Kendall Squares, which lie between Harvard University and the Massachusetts Institute of Technology campuses, and Lechmere, where numerous technology jobs are clustered. While a handful of clusters were able to attract more visits during some summer and fall months of 2020, overall, we only find one cluster on South Medford Main Street in Fig 3, where the mean year-on-year visits during the pandemic have been consistently higher (+198%) than before the pandemic. Upon closer examination, we learned that this is a cluster of small, largely Brazilian and Italian ethnic restaurants located in an otherwise dense residential fabric (photo available in Supp. Materials E in S1 File). One of the restaurants—Oasis—was a popular Brazilian buffet even before the pandemic; it appears to have successfully pivoted its operations to handle large volumes of take-out customers during the pandemic. The Somerville area has a relatively high proportion of Brazilian ethnic construction workers, whose work activities (along with essential businesses) were never halted as part of the shut-down order in 2020.

In 2019, the overall volume of trips rose from February to April, decreased over the summer, and peaked again in September—retail expenditure is typically lowest at the beginning of the year (after the holiday season and before taxes are due) and rises in the first few months, but declines during summer months when people go on holiday and the Boston area loses some of its student, staff, and faculty population (Fig 4) [40].

## 2.3. Analysis framework

A key methodological challenge to modeling how patronage differences across clusters may be affected by cluster attributes before and during the pandemic lies in treating trips between origins and destinations as a system, where origin-destination pairs are interdependent of each other. Linear regressions are inappropriate in this context since how many trips go to a

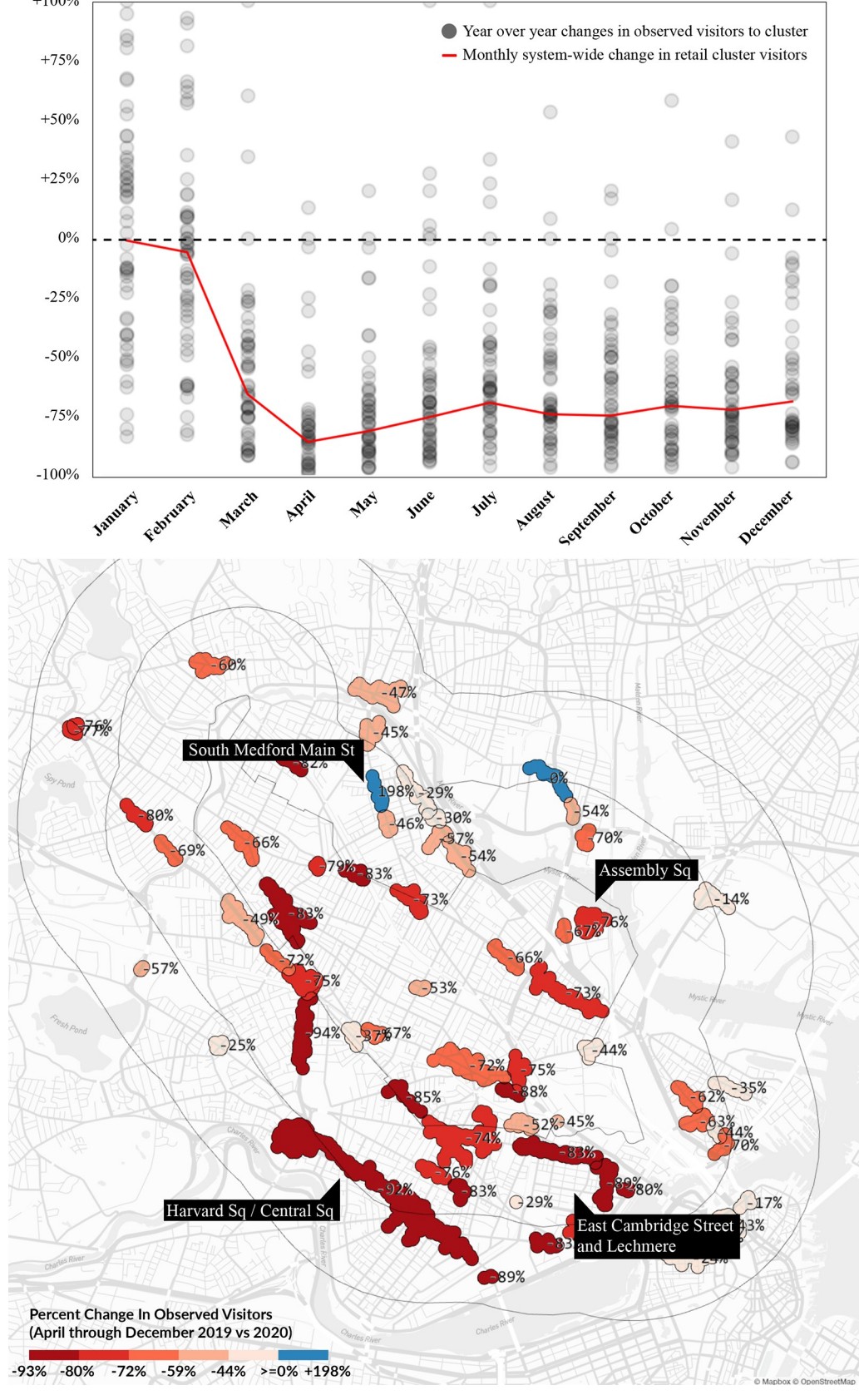

**Fig 3. 2020 change in observed visitors to amenity clusters compared to 2019 and a map of mean percent change in observed visitors to amenity clusters (April through December 2020 compared to 2019).** Base map and data from OpenStreetMap and OpenStreetMap Foundation.

particular cluster from each CBG depends directly on the availability and attributes of competing clusters, and on how many trips the competing clusters obtain as a result. Such interdependence clearly violates the IID assumption of ordinary least squares specification.

Since our application of examining amenity preferences is related to destination choice, we use a logistic choice model originally developed for analyzing factors that affect consumer patronage of shopping centers by Weisbrod et al. [13] and described in DiPasquale & Wheaton [12]. The model details are provided in Supp. Materials (D) in S1 File. Though discrete choice models typically assume individual-level choices, our data from SafeGraph contain aggregate trip volumes between CBGs and amenity clusters. This model enables aggregate choice estimation when the number of individual users for each alternative is large enough that it is reasonable to view them as shares or probabilities. Our data from SafeGraph includes thousands of individual trips each month, with a sufficiently large share of trips from each CBG to a number of surrounding amenity clusters, making CBG-based aggregation reasonable. Analogous, aggregated discrete choice models have been previously discussed by Berry [10] and Berry et al. [11].

While typical logistic choice models use a single alternative as the reference or baseline, observed destination choice data may not have a single common destination across individuals that can be considered as the baseline (as is the case for our analysis). Therefore, we construct a median cluster for trip-makers from each CBG based on the 'real' clusters to which trips were observed (e.g., a synthetic cluster $M$ with all median characteristics) and use that as the baseline to understand how other 'real' clusters are preferable (or not) to this synthetic cluster. Denoting the probability of a visitor from CBG $i$ to a cluster $j$ as $P_{ij}$ and noting a vector of characteristics at cluster $j$ as $X_j$, we write:

$$\ln(P_{ij}) - \ln(P_{iM}) = (X_j - X_M)\beta + \xi_j - \xi_M \tag{1}$$

To estimate choice preferences for aggregated shares, we construct the left-hand-side variable by taking logs of all the shares and then subtract the log-share of choice M from the log-shares of each of the other destination choices. This can be estimated with OLS, where the

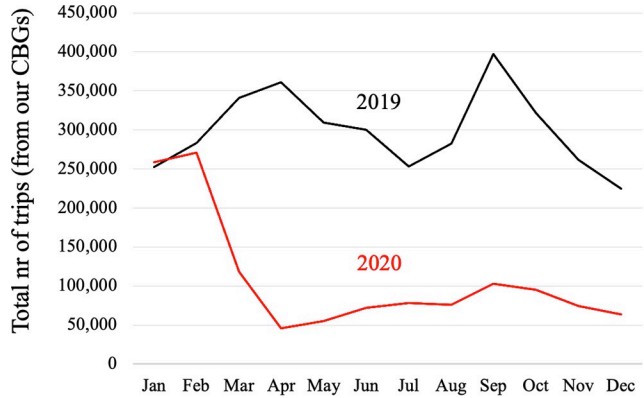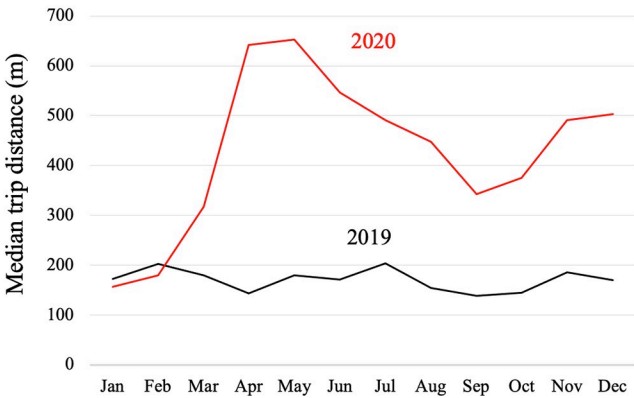

**Fig 4. 2020 changes in observed amenity-oriented trip volumes (left) and median trip distances (right) from CBGs in our study area, as compared to 2019.** Note: We observe trips from the CBG centroid to particular establishments in a cluster (rather than the cluster centroid), and compute the median distance from a CBG centroid to a cluster.

intercept becomes $\xi_M$ and the error terms are $\xi_j$:

$$\ln(s_j) - \ln(s_M) = \xi_M + (X_j - X_M)\beta + \xi_j \tag{2}$$

Centering choice analysis on shares of trips from specific block groups to amenity clusters constrains the interpretation of preference coefficients to a "typical" block group resident, rather than any specific resident. We also acknowledge that we are unable to observe intra-block group (or inter-consumer) heterogeneity due to the aggregate nature of the data available to us. In the analysis below, we are thus forced to limit our inference to Census block groups and summarize choices at the block group level as block group preferences (even though we fully understand that choices are made by individuals, not block groups).

We focus the analysis specifically on amenity clusters in Somerville, MA, but in order to enable trips to these clusters from all directions, we additionally include CBGs as trip origins within a one-kilometer buffer radius around Somerville (Fig 1). There is a total of 163 CBGs within the 1km buffer zone of Somerville.

In terms of destinations, there are 18 different amenity clusters, whose centroid falls within Somerville. In order to avoid an "edge effect" for destination choice, whereby trips to clusters within Somerville could only be analyzed from CBGs in Somerville, we include an additional 54 clusters that are within a 2km buffer area around Somerville. Each amenity cluster has a set of attributes that we hypothesize to affect visitor trips—total number of stores in the cluster, total store area in cluster, diversity of stores, land values in cluster (a proxy for how expensive a cluster may be), percent of large stores in a cluster, parking availability, availability of open recreational space, percent of stores that sell comparison goods, convenience goods, personal goods, and food and beverage services. The eleven cluster attributes included in our model are explained with descriptive statistics in Table 1. Median trip distances and total trip volumes across all CBGs over time are charted separately in Fig 4.

Cluster characteristics are constant during the observation periods in 2019, but vary during different months of 2020, since cluster characteristics reflect what is actually open and available to visitors at the time. Due to the shut-down order, a large portion of businesses were closed in 2020, consequently affecting cluster composition, size, and diversity. For instance, the median number of establishments in a cluster is 17 in 2019, but decreases to 10 between March-May

**Table 1. Descriptive summary of dataset using median values.**

| Variable [1] | Interpretation | 2019 | 2020 | | | | |
|---|---|---|---|---|---|---|---|
| | | | Feb | Mar-May | Jun | Jul-Nov | Dec |
| Nr establishments | Median nr of establishments in cluster | 17 | 17 | 10 | 16 | 16 | 16 |
| % Large | Share of establishments in cluster with Gross Floor Area > 40,000 sq.ft. | 3.2% | 3.2% | 0% | 1.5% | 1.5% | 1.5% |
| Land value | Average land value of cluster ($/sq.m.) per establishment. | 1059.0 | 1059.0 | | | | |
| Diversity | Simpson's Diversity Index of cluster by area of establishment types (ranges 0–1) | 0.6 | 0.6 | 0.1 | 0.7 | 0.6 | 0.6 |
| Parking | Number of parking spaces per establishment in cluster | 26 | 26 | 117 | 27 | 26 | 26 |
| Cluster area | Total area of establishments in cluster (thousands of sq.ft.) | 146.5 | 146.5 | 66.5 | 128.0 | 128.0 | 124.3 |
| % Comparison | Share of comparison retail (NAICS 441, 442, 443, 448, 451, 453) establishments in cluster | 12.6% | 12.6% | 0% | 14.2% | 14.2% | 14.3% |
| % Convenience | Share of convenience retail (NAICS 445, 447, 452, 456, 491) establishments in cluster | 10.0% | 10.0% | 18.2% | 10.0% | 10.0% | 11.1% |
| % F&B | Share of food and beverage (NAICS 722) establishments in cluster | 32.3% | 32.3% | 56.2% | 32.9% | 32.9% | 33.3% |
| % Personal | Share of personal care (NAICS 811, 812) establishments in cluster | 13.0% | 13.0% | 0% | 14.3% | 14.3% | 12.0% |
| Open space | Open public space, e.g. parklets, seating areas, etc. (sq.ft.). Source: OpenStreetMap. | 1448.0 | 1448.0 | | | | |

Note: We report the descriptive summary only for OD pairs that are less than 3 kilometers apart. The variables are cluster attributes and depend only on the destination cluster, so we summarize these distributions at the cluster level.

2020, and goes up to 16 in June, during the second phase of business reopening. Convenience businesses, such as grocery stores, pharmacies, convenience stores were kept open as "essential businesses" during the initial shut-down order; we consequently see their share of the cluster composition as significantly higher during March-May 2020 (18.2%) than in 2019 (10%). The share of comparison and personal stores, which describe the share of clothing, apparel and shoe stores and the share of beauty, health and repair businesses in clusters, drops to zero during March-May 2020 due to regulatory restrictions. Variables reflecting per amenity conditions, such as availability of parking per establishment, and business diversity reflect monthly opening conditions. The only variable that remains constant in 2020 is Open Space Area, which describes the extent of public space within each cluster.

The total number of trips made from all CBGs combined during different months, and the mean distance traveled to an amenity cluster diverge widely between 2019 and 2020 (Fig 4; S1 Table in S1 File). While trip volumes have been far below the 2019 levels throughout 2020, median trip distances (measured as network distances) have increased considerably—from a fairly stable median of ~200m in 2019 to over 600m in April and May and between 350-600m between June and December 2020. This suggests a lower disutility of distance among destination choice factors during the pandemic, which our model results also indicate below.

## 3. Results

We present model results for all block groups in Tables 2 and 3, illustrating how changing preferences towards both travel distance and cluster attributes have affected visits to amenity clusters in and around Somerville, MA in 2019 and 2020 respectively. In order to compare month-to-month shifts in coefficients in a regular year (2019) to those in a pandemic year (2020), we estimated separate models for each month in both years. Standard errors were calculated using the delta method, which has been found to be appropriate for use with large samples and is most accurate for well-conditioned data [41]. Some of the variables had to be scaled for the choice model (Supp. Materials C in S1 File), which are unscaled for effect interpretation below.

The models' adjusted $R^2$ fit varies from lowest in October (20% in Oct. 2019 and 9% in Oct. 2020) to highest in May (30% in May 2019 and 31% in May 2020) on both years—a relatively robust fit for a discrete choice model. The model fit is relatively consistent ($R^2$ 20–30%) during pre-COVID-19 months in both 2019 and 2020 but decreases during COVID-19 as travel behavior exhibits greater idiosyncrasies. We observe a decrease in $R^2$ in April compared to March 2020 (from 22.3% to 17.9%)—the first month following the public announcement of the pandemic, suggesting eccentricity in preferences as McKercher [27] and Fenichel et al. [26] also found for the periods following the SARS and H1N1 outbreaks. However, in May 2020, the fit actually increased to 30%, likely because our model only captures home-based trips and most workers were conducting their trips from home by May. Overall, the 2020 monthly models have a lower fit than 2019 models due to a significant decrease in trip quantity—which increased error variance—as well as added behavioral uncertainty during the pandemic. Since our dependent variable models the share of trips—or probabilities—to each destination cluster from available options compared to a reference cluster with median attributes—the signs and magnitudes of estimated model coefficients capture how specific cluster attributes have impacted their likelihood of being visited. Coefficients do not capture variations in trip volumes but rather shifts in destination preference.

The "raw" behavioral model coefficients reflect the log-odds of the independent variables on trip-making probability. Discrete choice models contain a scale parameter that is assumed

**Table 2. Model results for all CBGs, for trips up to 3km in length, in 2019.**

| | Trips to amenity clusters in Somerville (Aggregate models < 3kms) - 2019 | | | | | | | | | | | |
|---|---|---|---|---|---|---|---|---|---|---|---|---|
| | *Dependent variable*: log(number of visits) | | | | | | | | | | | |
| | Jan | Feb | Mar | Apr | May | Jun | Jul | Aug | Sep | Oct | Nov | Dec |
| **Constant** | 0.068** (0.031) | -0.006 (0.030) | 0.074** (0.031) | 0.033 (0.030) | 0.023 (0.029) | 0.040 (0.029) | 0.025 (0.031) | 0.051* (0.030) | 0.053* (0.032) | 0.057* (0.031) | 0.045 (0.030) | 0.064** (0.029) |
| **Distance** | -0.426*** (0.041) | -0.448*** (0.042) | -0.469*** (0.040) | -0.504*** (0.040) | -0.527*** (0.038) | -0.416*** (0.037) | -0.438*** (0.041) | -0.432*** (0.039) | -0.483*** (0.040) | -0.443*** (0.042) | -0.416*** (0.042) | -0.440*** (0.041) |
| **Nr Establishments** | -0.023 (0.097) | 0.093 (0.097) | 0.223** (0.101) | 0.011 (0.092) | 0.089 (0.091) | -0.099 (0.087) | 0.040 (0.101) | 0.095 (0.097) | -0.028 (0.102) | 0.002 (0.100) | -0.001 (0.094) | -0.072 (0.094) |
| **% Large** | 0.958* (0.510) | 1.722*** (0.643) | 2.578*** (0.621) | 2.601*** (0.606) | 2.089*** (0.575) | 1.161*** (0.435) | 1.743*** (0.624) | 2.058*** (0.496) | 1.101** (0.519) | 2.058*** (0.538) | 0.807 (0.534) | 2.074*** (0.542) |
| **Land Value** | -0.039 (0.041) | 0.035 (0.045) | -0.013 (0.042) | -0.018 (0.037) | -0.022 (0.037) | -0.086** (0.038) | -0.073* (0.039) | 0.004 (0.041) | 0.0003 (0.042) | -0.011 (0.042) | -0.041 (0.041) | -0.075* (0.04) |
| **Diversity** | 1.131*** (0.204) | 0.691*** (0.213) | 0.553*** (0.205) | 1.158*** (0.192) | 0.727*** (0.171) | 0.752*** (0.184) | 0.647*** (0.198) | 0.649*** (0.202) | 0.806*** (0.192) | 0.689*** (0.213) | 0.877*** (0.196) | 0.760*** (0.220) |
| **Cluster Area** | 0.120 (0.082) | 0.001 (0.086) | -0.092 (0.089) | 0.057 (0.079) | -0.070 (0.082) | 0.151** (0.074) | -0.021 (0.088) | -0.045 (0.085) | 0.146* (0.088) | 0.021 (0.088) | 0.081 (0.080) | 0.083 (0.080) |
| **Parking** | -0.001 (0.001) | -0.001 (0.001) | -0.002*** (0.001) | -0.0001 (0.001) | -0.002** (0.001) | -0.001* (0.0003) | -0.001 (0.001) | -0.002*** (0.001) | -0.002*** (0.001) | -0.002*** (0.001) | -0.001 (0.001) | -0.001** (0.001) |
| **% Comparison** | 0.998*** (0.284) | 0.544 (0.333) | 0.553* (0.297) | 0.476 (0.290) | 0.929*** (0.284) | 0.478* (0.248) | 0.962*** (0.324) | 0.899*** (0.268) | 1.063*** (0.273) | 0.780*** (0.290) | 0.960*** (0.264) | 0.977*** (0.281) |
| **% Convenience** | -0.628 (0.499) | 0.392 (0.523) | -0.559 (0.532) | -1.076** (0.483) | 0.198 (0.463) | -0.225 (0.433) | -0.118 (0.488) | 0.063 (0.497) | 0.767 (0.478) | -0.158 (0.480) | 0.076 (0.484) | -0.799* (0.484) |
| **% F&B** | 0.354 (0.281) | 0.389 (0.291) | 0.101 (0.277) | 0.349 (0.272) | 0.540** (0.243) | 0.302 (0.229) | 0.484* (0.279) | 0.047 (0.253) | 0.666** (0.263) | 0.696** (0.282) | 0.432 (0.277) | 0.998*** (0.287) |
| **% Entertainment** | -1.847 (1.312) | 2.563** (1.228) | 0.223 (1.214) | 0.085 (1.234) | 0.420 (0.967) | -1.232 (1.154) | 0.479 (1.109) | -0.963 (1.223) | 1.571 (1.099) | 0.141 (1.186) | 0.955 (1.007) | -1.146 (1.044) |
| **% Personal** | 1.120*** (0.422) | 0.411 (0.414) | 0.602 (0.405) | 0.802** (0.361) | 0.695** (0.349) | 0.551 (0.344) | 0.819** (0.395) | 0.360 (0.393) | 0.412 (0.396) | 0.628 (0.414) | 0.788** (0.367) | 1.007** (0.407) |
| **Open Space** | 0.012 (0.008) | 0.013 (0.008) | 0.014* (0.008) | 0.024*** (0.007) | 0.028*** (0.007) | 0.029*** (0.007) | 0.027*** (0.008) | 0.021*** (0.008) | 0.009 (0.007) | 0.018** (0.007) | 0.016** (0.007) | 0.008 (0.007) |
| **Observations** | 533 | 534 | 599 | 635 | 633 | 625 | 560 | 588 | 665 | 575 | 534 | 509 |
| **$R^2$** | 0.26 | 0.243 | 0.244 | 0.274 | 0.315 | 0.254 | 0.243 | 0.25 | 0.249 | 0.216 | 0.238 | 0.245 |
| **Adjusted $R^2$** | 0.242 | 0.224 | 0.227 | 0.259 | 0.3 | 0.238 | 0.225 | 0.233 | 0.234 | 0.198 | 0.219 | 0.225 |
| **Residual Std. Error** | 0.598 (df = 519) | 0.598 (df = 520) | 0.646 (df = 585) | 0.624 (df = 621) | 0.593 (df = 619) | 0.585 (df = 611) | 0.603 (df = 546) | 0.593 (df = 574) | 0.678 (df = 651) | 0.616 (df = 561) | 0.579 (df = 520) | 0.563 (df = 495) |
| **F Statistic** | 14.060*** (df = 13; 519) | 12.807*** (df = 13; 520) | 14.501*** (df = 13; 585) | 18.032*** (df = 13; 621) | 21.866*** (df = 13; 619) | 16.026*** (df = 13; 611) | 13.511*** (df = 13; 546) | 14.683*** (df = 13; 574) | 16.642*** (df = 13; 651) | 11.880*** (df = 13; 561) | 12.470*** (df = 13; 520) | 12.355*** (df = 13; 495) |

*Note*

*p<0.1

**p<0.05

***p<0.01

Standard errors in brackets.

to be unity for identification purposes. Therefore, it is not straightforward to directly compare raw coefficients across models, even for the same variable. One way to circumvent this issue is to use coefficient ratios. We use the distance variable as the denominator such that each coefficient ratio can be interpreted as the distance-equivalent impact on trip-making probability [42, 43]. For instance, the tradeoff between gaining access to additional establishments and

**Table 3. Model results for all CBGs, for trips up to 3km in length, in 2020.**

| | Jan | Feb | Mar | Apr | May | Jun | Jul | Aug | Sep | Oct | Nov | Dec |
|---|---|---|---|---|---|---|---|---|---|---|---|---|
| **Trips to amenity clusters in Somerville (Aggregate models < 3kms) - 2020** | | | | | | | | | | | | |
| *Dependent variable*: log(number of visits) | | | | | | | | | | | | |
| **Constant** | 0.065** (0.030) | 0.047 (0.033) | -0.015 (0.030) | -0.008 (0.027) | 0.014 (0.033) | 0.003 (0.030) | 0.020 (0.026) | 0.058** (0.026) | 0.072** (0.032) | 0.018 (0.036) | 0.013 (0.031) | 0.047 (0.032) |
| **Distance** | -0.396*** (0.041) | -0.435*** (0.045) | -0.383*** (0.049) | -0.309*** (0.060) | -0.400*** (0.059) | -0.227*** (0.050) | -0.255*** (0.045) | -0.266*** (0.045) | -0.223*** (0.049) | -0.251*** (0.063) | -0.312*** (0.056) | -0.200*** (0.052) |
| **Nr Establishments** | 0.044 (0.097) | 0.012 (0.110) | -0.130 (0.167) | -1.18* (0.61) | -1.791*** (0.563) | -0.020 (0.162) | 0.246** (0.108) | 0.056 (0.104) | 0.037 (0.114) | -0.054 (0.149) | 0.066 (0.108) | -0.017 (0.116) |
| **% Large** | 1.977*** (0.600) | 1.559** (0.671) | 0.117 (0.092) | 0.763 (0.607) | -0.446 (0.393) | 0.276 (0.215) | 0.920 (0.672) | 2.754*** (0.666) | 1.486** (0.695) | 0.760 (0.554) | 2.468*** (0.791) | 0.375 (0.585) |
| **Land Value** | -0.057 (0.040) | -0.076 (0.049) | 0.013 (0.047) | 0.116 (0.090) | -0.057 (0.055) | -0.170*** (0.060) | -0.074 (0.063) | -0.087** (0.043) | -0.127** (0.055) | -0.021 (0.071) | -0.035 (0.052) | -0.13* (0.073) |
| **Diversity** | 0.999*** (0.210) | 1.121*** (0.242) | -0.007 (0.400) | 0.051 (0.156) | -0.098 (0.183) | -0.002 (0.329) | 0.251 (0.242) | 0.341 (0.258) | 0.563** (0.267) | 0.385 (0.333) | 0.962*** (0.281) | 0.892** (0.361) |
| **Cluster Area** | -0.013 (0.084) | 0.052 (0.092) | 0.243 (0.160) | 1.163** (0.557) | 2.582*** (0.537) | 0.050 (0.144) | -0.108 (0.100) | -0.101 (0.093) | 0.003 (0.101) | 0.096 (0.121) | -0.0005 (0.097) | 0.049 (0.088) |
| **Parking** | -0.002** (0.001) | -0.001 (0.001) | -0.001 (0.001) | 0.0001 (0.0002) | 0.0001 (0.0002) | -0.001 (0.001) | -0.0005 (0.001) | -0.0002 (0.001) | -0.001* (0.0003) | -0.0003 (0.0004) | -0.002** (0.001) | -0.0003 (0.001) |
| **% Comparison** | 1.362*** (0.314) | 0.651* (0.349) | 0.419 (0.325) | -0.218 (0.825) | -0.095 (0.592) | 0.595 (0.367) | 0.354 (0.345) | -0.110 (0.316) | -0.114 (0.330) | 0.124 (0.438) | 0.738* (0.405) | 0.362 (0.361) |
| **% Convenience** | -0.168 (0.474) | -0.386 (0.579) | 0.654 (0.584) | 0.103 (0.243) | -0.054 (0.226) | -0.755 (0.573) | 1.036** (0.505) | -0.942* (0.503) | 0.259 (0.582) | 0.918 (0.733) | -0.109 (0.602) | -0.669 (0.589) |
| **% F&B** | 0.891*** (0.291) | 0.163 (0.351) | 0.951 (0.621) | | | 0.816 (0.527) | 0.181 (0.354) | -0.079 (0.344) | -0.138 (0.355) | -0.291 (0.439) | 0.657 (0.449) | -0.349 (0.407) |
| **% Entertainment** | 0.052 (1.197) | -1.881 (1.430) | 2.602** (1.230) | | | 4.003* (2.407) | 0.299 (2.893) | -2.858 (2.694) | 0.773 (3.527) | 1.126 (3.866) | 3.564 (4.375) | 2.206 (2.916) |
| **% Personal** | 0.851** (0.377) | 1.003** (0.426) | 2.153** (0.895) | 0.647 (0.434) | 1.124*** (0.384) | 2.031*** (0.597) | 0.367 (0.395) | 1.283*** (0.373) | 0.050 (0.420) | 0.617 (0.562) | 1.606*** (0.554) | 1.013** (0.447) |
| **Open Space** | 0.012 (0.007) | 0.022** (0.009) | 0.001 (0.009) | -0.007 (0.013) | -0.001 (0.011) | 0.008 (0.010) | 0.012 (0.009) | 0.020*** (0.007) | 0.022** (0.009) | 0.010 (0.011) | 0.007 (0.012) | 0.008 (0.010) |
| **Observations** | 568 | 513 | 332 | 171 | 182 | 222 | 261 | 264 | 285 | 253 | 230 | 216 |
| **R²** | 0.226 | 0.238 | 0.254 | 0.232 | 0.35 | 0.244 | 0.239 | 0.24 | 0.168 | 0.137 | 0.308 | 0.151 |
| **Adjusted R²** | 0.208 | 0.218 | 0.223 | 0.179 | 0.308 | 0.196 | 0.199 | 0.2 | 0.128 | 0.09 | 0.266 | 0.096 |
| **Residual Std. Error** | 0.591 (df = 554) | 0.636 (df = 499) | 0.467 (df = 318) | 0.324 (df = 159) | 0.382 (df = 170) | 0.395 (df = 208) | 0.380 (df = 247) | 0.386 (df = 250) | 0.478 (df = 271) | 0.510 (df = 239) | 0.423 (df = 216) | 0.411 (df = 202) |
| **F Statistic** | 12.431*** (df = 13; 554) | 11.997*** (df = 13; 499) | 8.327*** (df = 13; 318) | 4.377*** (df = 11; 159) | 8.314*** (df = 11; 170) | 5.155*** (df = 13; 208) | 5.974*** (df = 13; 247) | 6.073*** (df = 13; 250) | 4.215*** (df = 13; 271) | 2.921*** (df = 13; 239) | 7.387*** (df = 13; 216) | 2.765*** (df = 13; 202) |

*Note*

*p<0.1

**p<0.05

***p<0.01

Standard errors in brackets.

travel distance can be shown as a ratio of their de-transformed coefficients:

$$\frac{\beta_{Num\_Establishments}}{\beta_{Distance}} = -21.2$$

Using descaled coefficients (Table 4), we can interpret this to imply that increasing the number of businesses in a cluster by 10 reduced the perceived distance to that cluster by 21.2

**Table 4. Distance-equivalent effects of cluster attributes in February, May and November 2020.**

| Variable | Length equivalent interpretation | Feb 20 | May 20 | Nov 20 |
|---|---|---|---|---|
| | | m | m | m |
| Nr Establishments | Having 10 additional establishment in a cluster is perceived as | -2.8 | 447.8*** | -21.2 |
| % Large | A 1% increase in large establishments is perceived as: | -35.8** | 11.2 | -79.1*** |
| Land Value | A $100 /m2 increase in avg land value in is perceived as: | 17.5 | 14.3 | 11.2 |
| Diversity | A 1% increase in diversity is perceived as: | -25.8*** | 2.5 | -30.8*** |
| Cluster Area | A 10,000 m2 increase in cluster area is perceived as: | -12.0 | -645.5*** | 0.0 |
| Parking | Adding one parking space per business is perceived as: | 2.3 | 0.0 | 6.4** |
| % Comparison | A 1% increase in comparison-goods stores is perceived as: | -15.0* | 2.4 | -23.7* |
| % Convenience | A 1% increase in convenience-goods stores is perceived as: | 8.9 | 1.4 | 3.5 |
| % F&B | A 1% increase in F&B stores is perceived as: | -3.7 | | -21.1 |
| % Entertainment | A 1% increase in entertainment stores is perceived as: | 43.2 | | -114.2 |
| % Personal | A 1% increase in personal-goods stores is perceived as: | -23.1** | -28.1*** | -51.5*** |
| Open Space | A 100 m2 increase in open space is perceived as: | -50.6** | 2.5 | -22.4 |

*p<0.1

**p<0.05

***p<0.01.

meters on average (in November 2020). This interpretation approach is quite similar to elasticity, although a key difference is that we are measuring elasticity with respect to distance, not with respect to trip-making probability. The value of this approach is also underscored by the policy relevance of a distance-equivalent measure, which helps us understand which variables can be modified through public policies and to what extent they may impact trip-making by reducing perceived distances to clusters.

In order to interpret how behavioral responses to specific cluster attributes have shifted over time, we have charted a longitudinal trend line for each effect separately in Figs 5, 6 and 7, where 2020 effects are marked in red and 2019 effects in black. Each of the charts presents descaled distance-equivalent coefficients corresponding to the interpretations provided in Table 4 as well as in the title of each chart. We present the distance-equivalent effects with a flipped Y axis, so that a rise in a trend line signals stronger preference towards a given attribute (a more negative distance equivalence). Only the distance coefficient, which is used as a denominator for other coefficients, is presented in its raw form in Fig 5.

Shaded bands around coefficients indicate 95[th] percent confidence intervals. A white gap between any two confidence bands indicates coefficient ratios that are significantly different from expected levels during the pandemic year. Overall, confidence intervals are wider in 2020, indicating more instability in trip making during the pandemic. In the following, we describe the effects separately for sensitivity to distance (Fig 5), physical cluster attributes (Fig 6), and cluster business composition (Fig 7).

### 3.1 Change in sensitivity to distance

Consistent with transportation and travel behavior theory, the distance coefficient (Fig 5) is negatively related to cluster patronage—the farther away a cluster is, controlling for other covariates, the less likely people are to visit it. In 2019, its magnitude increases at the beginning of the year, suggesting an increased preference for destination proximity during the working months of February to May. This could be due to more trips on foot to local clusters with warmer weather and more trips by car to farther clusters during colder weather. At the same

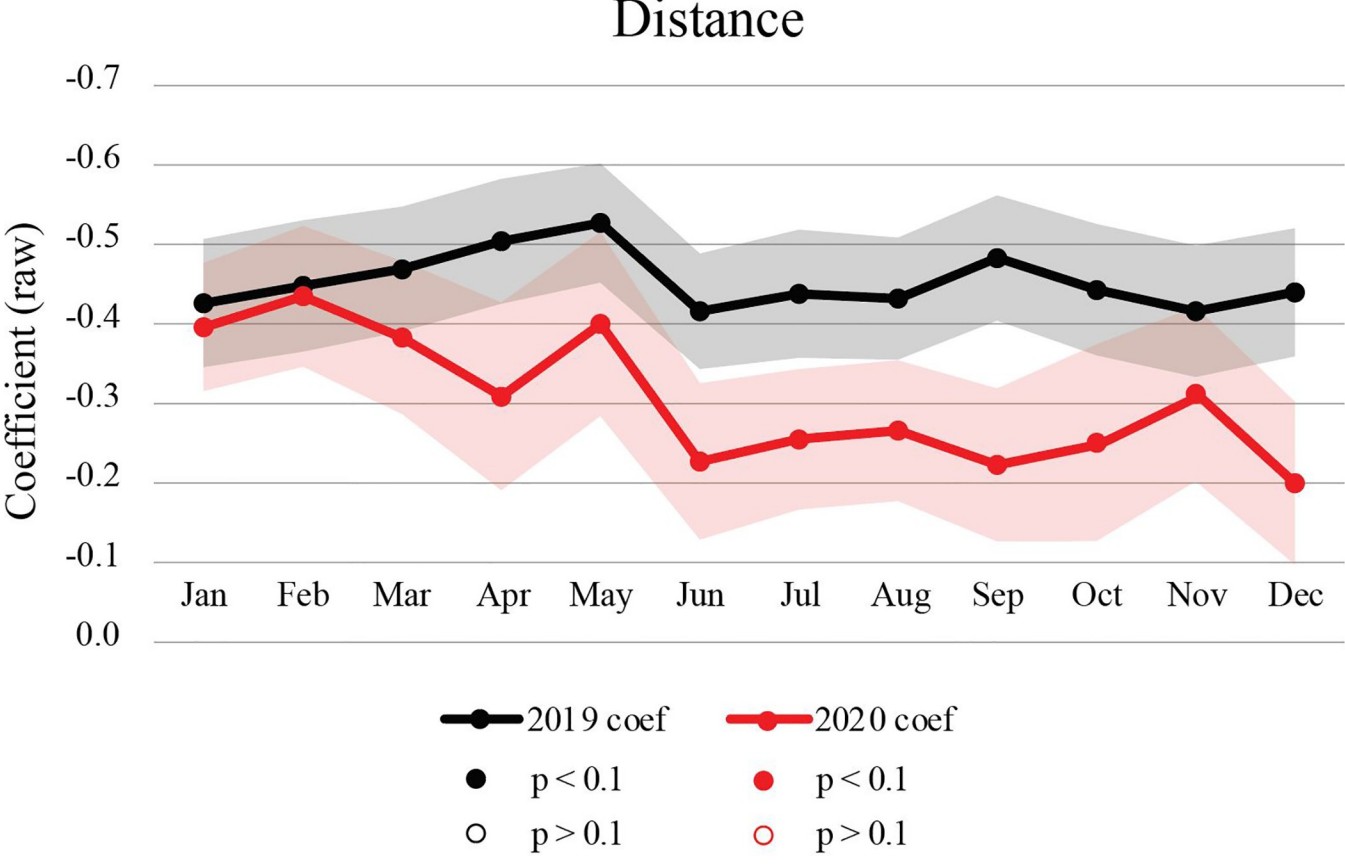

**Fig 5. Estimated preferences towards distance from origin CBG to destination cluster, describing 2019 and 2020 raw distance coefficients.**

time, we note a decrease in the 2019 trendline during summer months, possibly due to fewer work-related schedule constraints and more recreational trips. In 2020, we find the distance coefficient to be quite similar to the 2019 coefficient in January and February, but notably different starting from March 2020, when the pandemic began. In all of the following months, the relative importance of distance decreased, suggesting a higher willingness to travel further to access amenity clusters. While in 2019, willingness to travel further decreases with warmer spring weather, we find the opposite in 2020. Longer journey distances suggest higher destination selectivity during the pandemic—preferences towards clusters that offer particular qualities, not necessarily those that are closest. This may be supported by a rise in vehicular travel (as opposed to public transport or walking), less traffic congestion on roadways, and more flexible travel schedules (given a larger share of the population working from home). A lesser preference for destination proximity may be of concern to transportation planners from the City of Somerville who have been devising progressive mobility policies for years, placing greater emphasis on walking, biking and public transit ridership, all of which support shorter trips and assume individuals to be more sensitive to travel distance.

## 3.2 Preferences towards physical cluster attributes

Visitor preferences towards some of the physical characteristics of clusters notably shifted during the pandemic (Fig 6 and Table 4). For instance, in both February 2019 and 2020, a cluster with 10 additional establishments was perceived as being marginally closer (2.8m)—a desirable

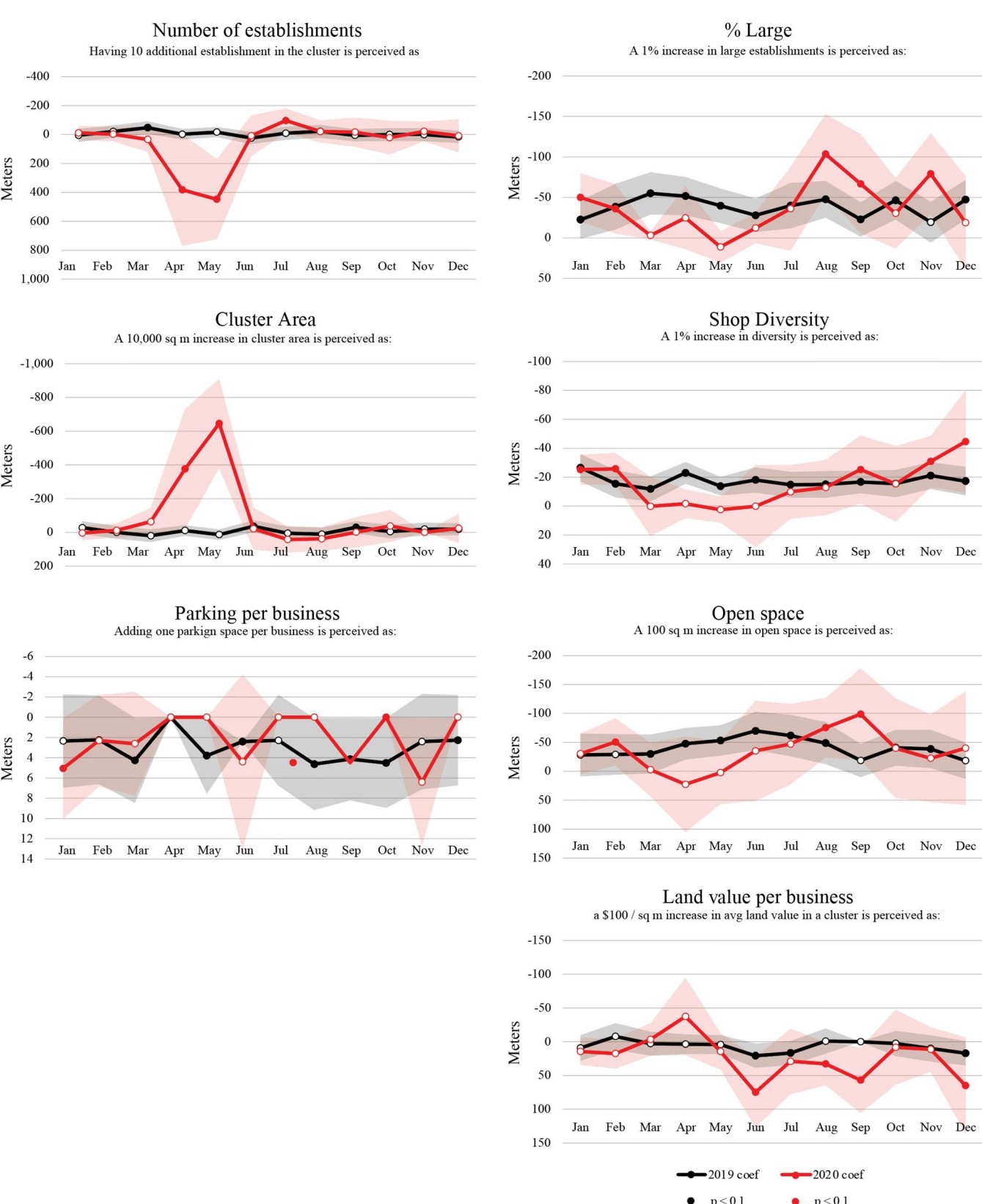

**Fig 6. Estimated preferences towards physical characteristics of destination clusters.** Trend lines represent distance-equivalent coefficients (ratio between coefficient and distance coefficient).

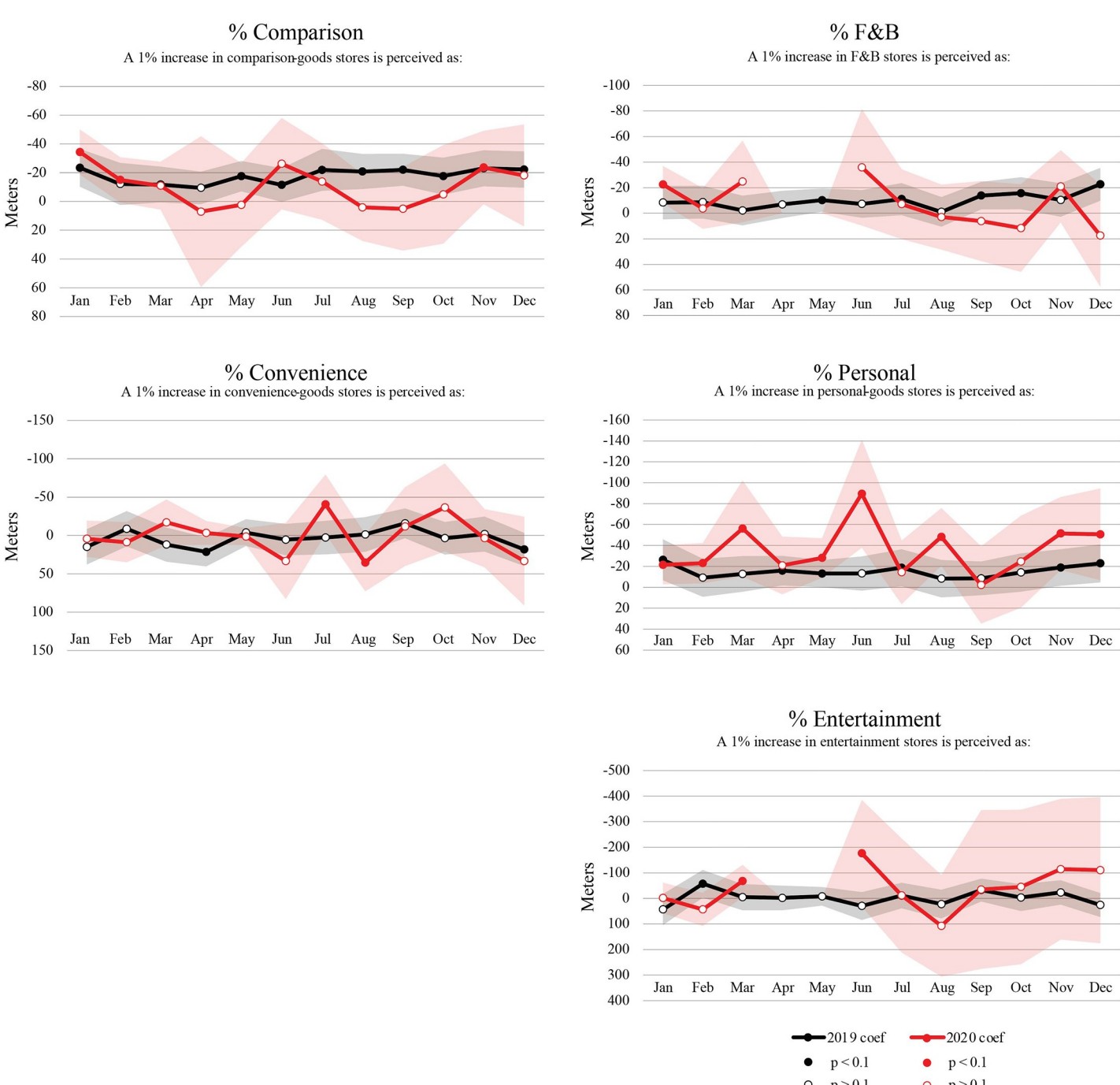

**Fig 7. Estimated preferences towards cluster business composition.** Trend lines represent distance-equivalent coefficients (ratio between coefficient and distance coefficient).

effect. In May 2020, on the other hand, during the height of the pandemic, a cluster with 10 additional establishments was seen as equivalent to 447.8 meters farther away from the visitor —clearly not a quality that people preferred at the time. By July 2020, having a higher number of establishments available at the destination was perceived as attractive again.

The 2019 data suggests that in a normal year, cluster preferences were positively related to the diversity of the business mix, the percent of large stores (covering at least 4,000 square

meters), the amount of open space in the cluster, and marginally to the number of stores in the cluster. At the start of the pandemic, the effect of each of these attributes shifted considerably downward, flipping signs in several instances. In April and May, for instance—the first full months of the pandemic—the diversity of the cluster had no impact on patronage, the presence of open gathering areas was perceived as insignificant or even negative. Results in Table 4 suggest that in May 2020, having an additional 100m$^2$ of open space was perceived equivalent to the cluster being 2.5 meters farther away—a quality that was being avoided. Having more establishments was also perceived as negative rather than positive during that month. At the same time, we note a preference increase towards cluster square footage in May 2020—a 10,000m$^2$ increase in cluster size was comparable to the cluster being 645 meters closer. This preference increase (which is close to zero in a normal year) speaks to the surge towards large grocery stores, general stores and home improvement stores in the early phase of the pandemic. Similarly, preferences towards the percent of large stores increased in April (height of the pandemic), but were down in May, after the initial rush on essential goods for sheltering at home. Our data suggests that in May 2020, people were avoiding big-box clusters (e.g. Assembly Square)—bigger stores may have been perceived as more hazardous due to the aerosol transmission of COVID-19. Albeit notable year-on-year differences among these coefficients between March and August, each of these variables started converging back to expected levels at the end of summer 2020. Even though absolute trip volumes in fall 2020 stayed far below the 2019 levels (Fig 3), among the stores that were open, patrons' preferences for cluster diversity, number of stores, cluster area, number of establishments, and the percent of large stores had largely shifted back to expected levels by September.

Parking and land values add interesting nuance to behavioral dynamics. The presence of parking spaces appears to have no significant effect on cluster visits during both a normal year and a pandemic year. Though parking coefficients were slightly higher during spring and summer 2020 than 2019, they were still perceived as negative (positive distance equivalence) and insignificant, suggesting that cluster visits are not influenced by parking availability during both a regular and a pandemic year. This offers some empirical support for the City of Somerville's progressive efforts of adding bike lanes and converting more curb-side parking spaces to outdoor dining spaces. This is also supported by the positive and significant "open space" effect from March to August during a normal year (as well as August and September 2020), which suggests that more open space—in the form of parklets, public squares, tables and benches—typically relates to more cluster visits during warmer months of the year. However, we acknowledge that our metric for parking was captured per business, making it dependent on business closures and thus imperfect for capturing preference effects conclusively.

Finally, land values—our proxy variable for the relative expensiveness of the cluster—have no significant effect on cluster visits during normal times. We note a slightly increased preference for patronizing more expensive locations in April 2020, which subsequently turns into a decrease and a preference for less expensive destinations after May 2020. The shift is only distinguishable from the 2019 levels in September 2020. Similar to Chetty et al. [36], this could signal higher levels of in-person shopping activity by lower income residents, who had to work and shop in person again by September 2020, compared to higher income residents with more means to shop online and work from home.

## 3.3 Preferences towards business composition

Cluster composition coefficients in Fig 7 illustrate how the proportional share of different types of stores can affect cluster visits. Stores selling comparison goods include shoes, clothing, apparel, hobby, music and book stores (see Table 1 for their NAICS codes) which patrons

often compare from store to store before making a purchase [44]. In 2019, a higher share of comparison-good stores relates to higher trip likelihoods and more customer visits per cluster. The positive effect of comparison stores on patronage is also supported by research on shopping center visits [45–47]. For instance, in a survey of 1,200 individuals in six malls in the U.S., Bloch and his colleagues found that visits to shopping areas without buying plans or with anticipated future buying plans constituted 62% of all trips [48]. Our Table 4 suggests that in February, a 1% increase in comparison-goods stores is perceived as equivalent to the cluster being 15m closer. The proportion of food and beverage businesses and personal-care stores in a cluster also have positive effects on destination choice in 2019, though with fewer months exhibiting statistically significant effects. Preferences shift somewhat during different phases of a normal year, with the positive impacts of food and beverage and personal-care stores rising towards the holiday season at the end of the year.

During the pandemic, we find lower coefficients for the percentage of comparison and food and beverage stores during most months. The month of June presents an exception, when those food and beverage, personal care and entertainment establishments that were open, show a greater positive effect than in 2019. This likely illustrates an initial increase of visits to such establishments when they could be visited again as part of Phase One re-openings in June 2020. Given that relatively few overall businesses were able to operate in June, those food and beverage and personal care stores that were open automatically constituted a higher overall percent share of the cluster. Interestingly, a spike in preference towards personal service business is observed in June 2020, when these businesses revived as part of Phase 2 reopening— this suggests taking care of "deferred maintenance"—getting haircuts, visiting beauty salons or repair services after the initial shutdown period. This is the only effect where the gap between 2019 and 2020 coefficients exceeds the 95% confidence band overlap. For all other cluster composition effects, our model is unable to confirm a significant difference in preferences compared to the prior year.

In summary, while preferences towards clusters' business composition had no significant difference between 2019 and 2020—which may be in part due to the closure of many businesses across categories—pandemic year preferences towards some of the physical cluster attributes did significantly shift beyond expected levels during the pandemic. During the first two months of the pandemic (April and June 2020), people were more selective in destination choices—they were more willing to travel further to larger clusters (in terms of square footage) with essential stores. Shop diversity and the number of individual stores played a significantly smaller role at the start of the pandemic than during the same months a year before. This fits with our exploratory establishment-level findings, where we observed that grocery stores, hardware and home improvement stores (which are typically found in larger amenity clusters) experience the highest number of visits relative to other open store types and the smallest overall decrease in visits during the early phase of the pandemic. By the end of summer 2020, all coefficients started converging back towards expected levels within 95% confidence intervals. While the quantity of trips made to amenity clusters still remained far lower than the pre-pandemic levels (Fig 3), preferences towards specific cluster characteristics appear to be converging back to pre-pandemic levels by fall 2020.

It is important to note that cluster characteristics themselves are not the sole driving force behind decisions to patronize business clusters. Behavior is driven by forces that are both internal and external to individuals. Yet for the purposes of understanding the characteristics that local governments and businesses themselves can help to steer and mitigate, these findings offer unique insights in consumer behavior and travel behavior during this unprecedented pandemic.

## 4. Scenario predictions

In March 2021, the COVID-19 pandemic is still ongoing. The federal government and states are rushing to deliver the vaccine as new virus variants along with rising infection numbers spread across the US. While some local and state governments are maintaining business restrictions to prevent the spread of the virus, many have started to reopen the economy to allow devastated businesses to gain income and patrons to gain access to goods and services. While we cannot tell whether new business restrictions could be in order, the above findings allow us to explore the impacts that different regulatory scenarios might have on amenity patronage.

Using behavioral coefficients from November 2020, when the 14-day moving average of newly detected infections in Somerville was around 35 per day (per 100,000 residents), we predict how visits to clusters might have changed with different regulatory scenarios, while taking into account that consumer behavior was no longer the same as it was in the pandemic's early phase in April, but indeed had evolved to the November 2020 stage. To do this, we use the behavioral coefficients estimated for November 2020 to model two potential scenarios. First, using observed November 2020 visits as a baseline, we examine how cluster visits could have been affected if the same business restrictions that were introduced in April 2020 would have been implemented again, but assuming that behavioral preferences would have remained similar to November 2020 levels. Second, we look at how cluster visits might have changed if all businesses were re-opened to pre-pandemic levels (i.e., as in February 2020), but again assuming consumer preferences from November 2020.

These scenarios are depicted in Fig 8. The upper left chart (a) illustrates the observed change in trips in April 2020, at the beginning of the pandemic, compared to expected visits according to the same month a year before. The "expected" trips are based on the number of trips in the same month in the previous year, adjusted for annual shifts between February 2019 and February 2020. Across all clusters combined, trips fell from 359,575 expected for April 2020 to an actual 45,975 in April 2020 (-87%). While some clusters (e.g., Medford Main street) lost "only" 19% of their visits that month, many others, including the largest cluster at Harvard Square—Central Square, lost over 90% of their visits. This represents the confluence of a regulatory shut-down, especially cautious consumer behavior at the start of the pandemic, and a lower population in residence (as universities sent students home in March). The upper right chart (b) illustrates the estimated effects of the same kind of shutdown in November 2020 (Scenario 1). The baseline for percent change is the observed November 2020 trip count (we only count trips less than 3 kilometers in length from our selection of CBGs in each figure and scenario for consistency). Since these visits were already considerably lower than a "normal" November a year earlier, we estimate a smaller adverse effect from a new shut-down—an 8% overall reduction in visits compared to the observed visits in November 2020. However, the impacts of a new shutdown would have varied across clusters. For instance, Davis Square would have lost 39% of its visits that month and Assembly Row 21%. Interestingly, we find that some clusters would have also gained from the shut-down as travel behavior would shift from the closed clusters to those clusters where essential stores (convenience goods, F&B) are still open (e.g., 11% gain in the largest Harvard Sq–Central Sq cluster). But compared to the total volume of trips expected in a normal November (2019) across all clusters (266,400 trips) the new shut-down would still produce very low patronage (68,175 trips)—a 74% decrease compared to the year before. The more limited severity of the impacts witnessed as compared to the April 2020 lockdown is thus due to both the on-going business restrictions in place in November 2020 and cautious consumer behavior.

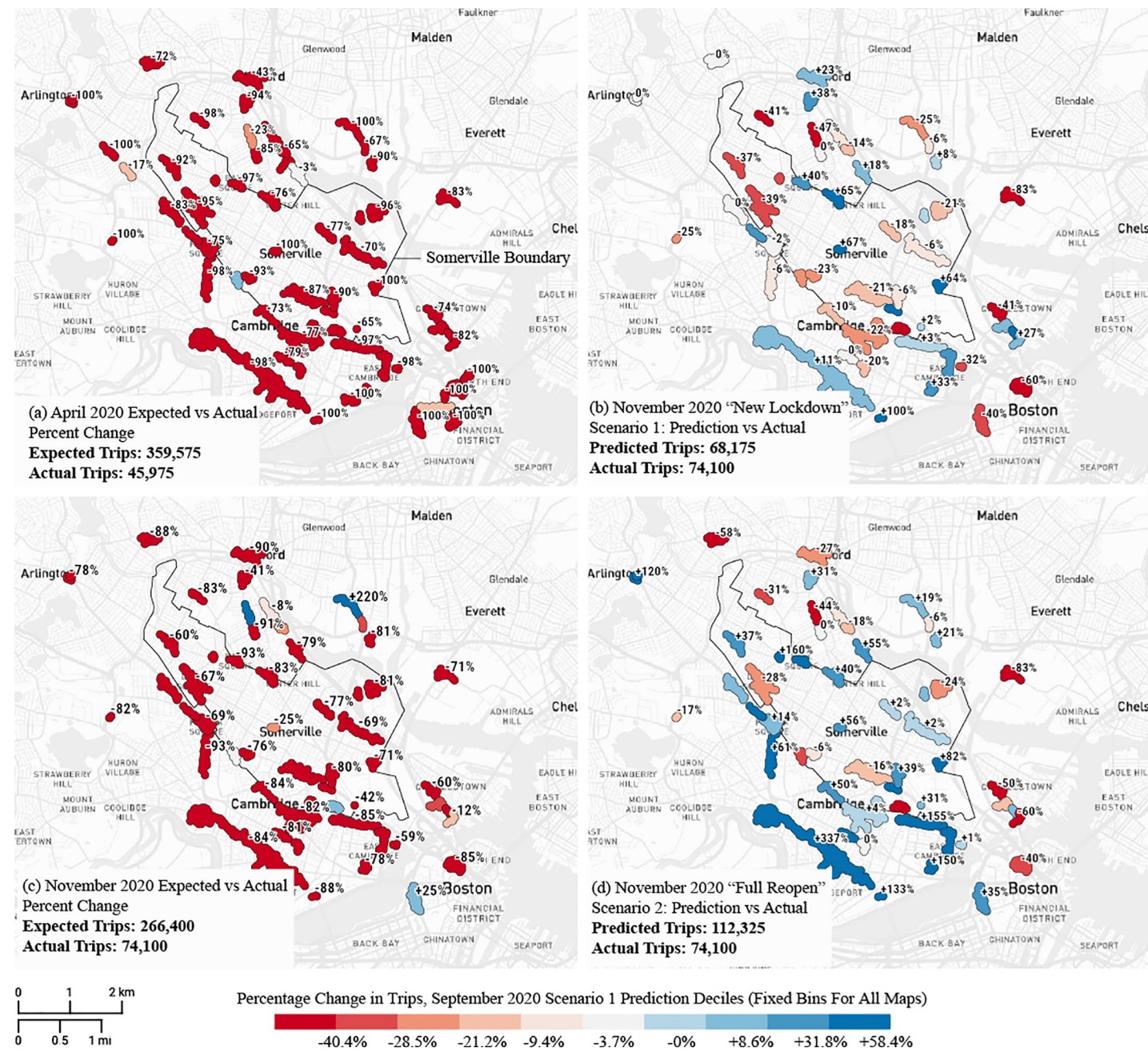

**Fig 8. Scenario analysis.** (a) Observed percentage change in visits in April 2020, during the business shut-down order, compared to expected visits based on April 2019 (adjusted with expected year-on-year change based on February differences between 2019 and 2020). (b) (Scenario 1) Estimated percentage decrease in cluster visits assuming a similar shut-down order as in April, but assuming November 2020 behavioral coefficients to persist. (c) Observed percentage change in cluster visits in November 2020 compared to expected visits based on November 2019. (d) (Scenario 2) Estimated percentage change in cluster visits with a full re-opening of all businesses back to the February 2020 level, compared to November 2020 observed baseline values. Base map and data from OpenStreetMap and OpenStreetMap Foundation.

How would a full reopening of all businesses (Scenario 2) have affected amenity visits using November 2020 as a baseline? The bottom left chart (c) illustrates the percent difference in actual cluster visits in November 2020 compared to expected visits in November 2020. Trips were down -72% compared to expected levels based on the year before. The bottom right chart

(d) estimates how a full business reopening would have changed visits compared to observed visits in November 2020. As expected, we forecast a considerable increase in trips from 74,100 observed in November 2020 to 112,325 trips with all business operational (as in February 2020). A full business reopening in November 2020 would have resulted in a 52% increase in visits across all clusters compared to observed visits in November 2020. But that still would not be enough to return amenity visits close to an expected level based on November 2019. Amenity-oriented trip behavior remained cautious in November 2020 and less frequent compared to the year before. With all businesses open, the total number of trips across all clusters (112,325) would have still been 58% lower than expected visits based on 2019 data—a big increase from the actual November 2020 situation, but far short of a normal November pattern.

## 5. Discussion

While both the State of Massachusetts and City of Somerville business shut-down orders have had a significant impact on reducing visits to amenity clusters, our consumer behavior analysis suggests that behavioral choices have also had a significant impact during the pandemic. Even among the stores that have remained open, visits have been significantly down. Visitor behavior was most cautious in April and May 2020, at the start of the pandemic. While the overall volume of trips to amenity clusters has slowly grown since, it still remained around 70% below 2019 levels in November and December 2020.

Despite making significantly fewer trips to amenity clusters, people's 2020 preference coefficients towards specific cluster attributes ultimately proved similar to 2019 preference coefficients, within 95% confidence intervals. Out of the 13 cluster attributes we analyzed, we found significant preference differences for only five attributes at some points during the pandemic. Perhaps most significantly, dislike for travel distance has decreased and people are undertaking longer journeys to access amenities. In the early phase of the pandemic (April-June 2020), people shied away from clusters with more businesses, but were drawn to clusters with a larger area—this indicates a preference towards large essential stores, such as grocery stores and home improvement stores that received disproportionately high visits during these months. In terms of preferences towards clusters' business composition, the only significant difference was observed in June 2020, when personal-goods businesses were particularly in demand—likely due to a backlog in personal services, such as haircuts, dental appointments, repair appointments etc. after the initial shutdown. These trends coincide with a sudden increase in e-commerce during the first few months of the pandemic.

Our scenario analysis for November 2020 suggested that even if businesses were opened back to pre-pandemic (February) levels, visits would still have remained 58% below expected levels based on prior year's data due to the lower patronage frequency on behalf of customers. This has implications for both controlling the spread of the pandemic and for the well-being of businesses.

From a public health standpoint, our findings suggest that highly cautious public policy championed by the City of Somerville has been only partly responsible for the reduction in trips taken by citizens during the pandemic. A big contributor to trip reduction is the behavior of Somerville residents, who have changed their travel patterns to amenity clusters, significantly reducing trip frequency throughout the pandemic and exhibited higher destination selectivity in the first half year of the pandemic. From a business perspective, this has brought along a painful loss of visits from which it will be difficult to recover in the short run. But even later in the year, if the city and state had allowed all business to return to the pre-pandemic levels, visits across all amenity clusters would still have remained low overall. Unlike regulatory

changes, which can be implemented relatively fast, rebuilding consumer confidence and habits could take much longer [30]. Therefore, more financial support, both federal and local, as well as policy innovation to curb rent burdens and business costs is likely going to be needed to ensure that businesses can weather a continued lack of customers over the upcoming months.

There are limitations to the SafeGraph mobile phone data used for our model, regarding how trips are counted, true trip origins, and multipurpose trip chaining. SafeGraph data report a minimum of four trips between any given destination and Census block group, suggesting a floor function in the data. We addressed this by only including a minimum of five trips between a CBG and cluster. Further, as the SafeGraph data retain the anonymity of the users that it tracks, it does not chain 'visits' across multiple destinations. Our analysis was thus unable to differentiate visits that were made by a user to multiple amenities during the same trip. We acknowledge this limitation and expect that the "number of businesses" and "cluster area" effects could be partly inflated due to trip chaining. SafeGraph data also do not indicate the time at which the person left the origin CBG, and could potentially include trips made from non-home origins. This is partially mitigated by the fact that Somerville is largely a residential town, with limited job access around the amenity clusters we examined. Even though the model cannot explicitly differentiate personal trips from trips performed by food-delivery services, both are likely included since a return journey by delivery personnel would produce a similar GPS trace as a personal trip to a cluster. While SafeGraph data does provide a voluminous overview of mobility and trip activity, these limitations do impact its fidelity.

Additionally, we modeled store closures by switching on/off specific sectors (based on NAICS 3-digit codes) for each month of the lockdown. This did not account for partial openings, such as private gyms or spas hosting by-appointment visits, or outdoor dining and alternative service adaptations.

There is an opportunity to expand this analysis beyond the borders of Somerville, to explore whether similar behavioral trends can be found in surrounding municipalities and elsewhere in the country. We anticipate that not all municipalities would see similar returns to pre-pandemic behaviors, and we would expect that the behavioral arc might be different, depending on different timings of the pandemic's arrival in different geographic regions of the country. The findings could also be extrapolated to revenue to offer more nuanced insights into purchasing behavior in addition to patronage behavior. It is possible that individuals are making fewer trips, yet purchasing more goods during each trip—a scenario that is likely to differ across different types of businesses. Examining changing travel behavior for amenity-oriented trips on a shorter timeline, using mobile phone data and a type of discrete choice model suited for aggregate-level analysis can offer policymakers novel opportunities to make informed healthcare and economic decisions.

## Supporting information

**S1 Dataset.**
(CSV)

**S1 Fig. Amenity cluster on Main Street in South Medford.** Amenity cluster on Main Street in South Medford, featuring a popular Brazilian buffer restaurant "Oasis", which featured a 198% growth in visits between April-December 2020, compared to the same months the year before. (Image source: authors).
(TIF)

**S1 File. Supplementary materials.**
(DOCX)

## Author Contributions

**Conceptualization:** Andres Sevtsuk, Rounaq Basu, Jorrit de Jong.

**Data curation:** Dylan Halpern, Kloe Ng.

**Formal analysis:** Andres Sevtsuk, Annie Hudson, Dylan Halpern, Rounaq Basu, Kloe Ng.

**Funding acquisition:** Andres Sevtsuk, Jorrit de Jong.

**Investigation:** Andres Sevtsuk, Annie Hudson, Dylan Halpern, Kloe Ng.

**Methodology:** Andres Sevtsuk, Rounaq Basu.

**Project administration:** Annie Hudson.

**Software:** Dylan Halpern.

**Supervision:** Andres Sevtsuk.

**Visualization:** Dylan Halpern.

**Writing – original draft:** Andres Sevtsuk, Annie Hudson.

**Writing – review & editing:** Andres Sevtsuk, Annie Hudson, Dylan Halpern, Rounaq Basu.

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
