## [Decision Letter · Decision Letter 0]

17 Feb 2021

PONE-D-20-35811

Accessing urban amenities during COVID-19: travel behavior changes and future outlooks for business clusters in Somerville MA.

PLOS ONE

Dear Dr. Sevtsuk,

Thank you for submitting your manuscript to PLOS ONE. After careful consideration, we feel that it has merit but does not fully meet PLOS ONE’s publication criteria as it currently stands. Therefore, we invite you to submit a revised version of the manuscript that addresses the points raised during the review process.

I have received two reviews. Both reviewers like the data and the research question. One reviewer does not think that you can answer your research questions and recommend rejection with the aggregate data you have. This reviewer thinks you need individual-level data to answer your research question. The second reviewer is more positive and recommends a minor revision. After reading the manuscript and the comments, I think that the negative reviewers raise valid points. Still, discrete choice models using aggregate data are useful in many situations, which may be one of those situations. However, it would be best if you made this case clearly in the manuscript.

I would like in your Analysis Framework section to do a better job motivating your empirical strategy, clearly explaining for a general reader why your method is valid to answer your research question. You describe your model in the appendix, but it is a general textbook-like description. I am looking more to explain why your empirical strategy is the best for your research questions, given the data restrictions. You should also add caveats to your discussion about data limitations.

There are several other points raise by the reviewers that I would like you to address. Please submit your revised manuscript by Apr 03 2021 11:59PM. If you will need more time than this to complete your revisions, please reply to this message or contact the journal office at plosone@plos.org. Please include the following items when submitting your revised manuscript:

We look forward to receiving your revised manuscript.

Kind regards,

Gabriel A. Picone

Academic Editor

PLOS ONE

Journal Requirements:

2. Please include your tables as part of your main manuscript and remove the individual files. Please note that supplementary tables (should remain/ be uploaded) as separate "supporting information" files

3.We note that Figure(s) 1, 2 and 6 in your submission contain map images which may be copyrighted. All PLOS content is published under the Creative Commons Attribution License (CC BY 4.0), which means that the manuscript, images, and Supporting Information files will be freely available online, and any third party is permitted to access, download, copy, distribute, and use these materials in any way, even commercially, with proper attribution. For these reasons, we cannot publish previously copyrighted maps or satellite images created using proprietary data, such as Google software (Google Maps, Street View, and Earth). For more information, see our copyright guidelines: http://journals.plos.org/plosone/s/licenses-and-copyright.

a)  You may seek permission from the original copyright holder of Figure(s) 1, 2 and 6 to publish the content specifically under the CC BY 4.0 license. 

4. Please upload a new copy of Figure 2 as the detail is not clear. Please follow the link for more information: https://blogs.plos.org/plos/2019/06/looking-good-tips-for-creating-your-plos-figures-graphics/" https://blogs.plos.org/plos/2019/06/looking-good-tips-for-creating-your-plos-figures-graphics/

6.We note that the grant information you provided in the ‘Funding Information’ and ‘Financial Disclosure’ sections do not match.

Reviewers' comments:

Reviewer's Responses to Questions

**Comments to the Author**

1. Is the manuscript technically sound, and do the data support the conclusions?

Reviewer #1: Yes

Reviewer #2: Partly

2. Has the statistical analysis been performed appropriately and rigorously? 

Reviewer #1: Yes

Reviewer #2: N/A

3. Have the authors made all data underlying the findings in their manuscript fully available?

Reviewer #1: Yes

Reviewer #2: No

4. Is the manuscript presented in an intelligible fashion and written in standard English?

Reviewer #1: Yes

Reviewer #2: Yes

5. Review Comments to the Author

Reviewer #1: This paper contributes to our understanding of how consumer choice and policies impact visitation during the COVID-19 pandemic. The authors use a discrete choice method to analyze cell phone visitation data and find that consumer patterns have changed over the course of the pandemic relative to last year. The authors use their model to simulate visitation patterns under a new “lockdown” and a “full reopen” scenario to show that changes in behavioral patterns imply that visitation would remain low even if businesses were fully opened.

Contribution:

The paper is generally well written and clearly organized. The paper contributes an important and underappreciated point that the removal of public health policies to mitigate disease spread would not simply return life to pre-pandemic times. Consumer preferences and risk aversion influence decisions to patronize businesses and will continue to do so as long as the threat of infection persists.

While I agree with the overall approach of the analysis, I have several comments that may improve the paper. I try to organize my comments by theme:

Comments about the methods:

How are the clusters determined? The methods do not describe this in enough detail. It sounds like the authors are familiar with the area. Did they know which businesses to cluster? Are there known boundaries of these clusters and the safegraph poi centroid is intersected with the known boundaries?

Clustering visits may overestimate visits as the same device may enter multiple POIs in a cluster. Do you address this in any way? If not, this should be discussed in the limitations.

Why were the monthly aggregates chosen rather than the weekly patterns? Why were only certain months chosen when data is available from Jan 2019?

It sounds like trips were limited to only those from census block groups from within 2km. What is the rationale for 2km? How sensitive are the results to this choice? Were all other trips omitted from the dataset? Wouldn’t this impact the regression results if, for instance, trips from greater distances occur more frequently on the weekends?

The description of the empirical model should be in the main text and requires additional explanation. While this sounds like a standard model in some fields, PLOS One is interdisciplinary, and many won’t be familiar with the model. Equation 1 looks like a logistic regression, but the model description suggests that equation 4 is the one being estimated with OLS.

In either case, there are visits from the same CBGs going to different clusters. Wouldn’t that violate the iid assumption? It seems like clustering of standard errors is necessary, but without a better understanding of the model, I cannot recommend a specific structure.

Comments about the results:

Table 1 results: Land value switches to a negative sign. This may be capturing the effects documented by Chetty et al. (2020) showing that higher-income areas are acting more risk-averse and likely have more capacity to avoid visiting stores.

The distance results by income in Appendix T2 may be capturing sorting behavior. People with higher income may choose to live near the amenity clusters so as to avoid travel costs. Similarly, lower-income populations may not be able to afford to live near the amenity clusters and are forced to travel longer distances. Attributing this to preference seems misleading.

The authors provide no interpretation of the regression coefficients. Are they marginal effects? Is .3 large or small? The comparison over time is helpful but please provide an interpretation of the coefficients.

Comments about the discussion:

The first paragraph of the discussion claims that their model provides unique insights to policymakers about behavioral change and the effects of policies on visitation to inform programs to support businesses. Couldn’t we just look at the visitation data in real-time to measure how actual visitation is change and design compensation schemes to react to actual visitation? I believe the authors are trying to claim that they can produce counterfactual simulations with their model. This is an important point and contribution of the model. However, their model only implicitly captures COVID-19 risk by estimating the model in different months. Would including risk attributes that relate to COVID transmission enable further parsing of the effects? For example, could you calculate occupancy density a la Benzell et al. (2020) using the square footage and visitation information? Could you use variation in cases over time to help attribute behavioral changes to risk aversion? I recognize that this would raise endogeneity issues, but it would help address the benefits of suppressing the virus to businesses.

The authors acknowledge in the discussion that partial closures or capacity limits are not explicitly considered in their analysis. The implication of this omission is that policy/regulation is mistaken for behavioral change. Have these regulations changed over time in Somerville? If so, you may be able to exploit this variation to parse out how much is regulation versus consumer choice.

How generalizable are the results? The study area is in close proximity to several universities. I would assume the area is relatively high income and high education.

Comments about presentation:

The variable labels in Table 1 are hard to read (they look like code names). Can you convert them to something more readable?

The graphical presentation of the results is an effective visual for communicating the results. However, there is no indication of estimate uncertainty (other than the p-value significance). 95% confidence intervals on the estimates would convey the information in the graphic. The points could be slightly shifted left and right so the plot elements do not overlap. The lines connecting the point estimates could be removed to minimize clutter since they do not convey much information.

Along these lines, the authors ask the reader to visually compare the results across time, but do not provide a statistical comparison. The standard hypothesis tests reported in the table pose a null hypothesis that the coefficient is 0. We are interested in how the coefficient differs over time (relative to last year). The authors choose to estimate the models for each year-month separately, which is fine. Estimating them in a single model with dummy variables would provide the coefficients (and hypothesis tests) that would address my comment. Standard errors should certainly be clustered by year-month in that case.

Report standard errors rather than p-values in Table 1.

Line 870: I believe mean should be median.

Line 524 describes the upper left panel of Fig 6 as comparing April visits in 2020 to 2019. However, the figure indicates a comparison of observed trips to expected. Is expected a model prediction? If so, I think the text needs to be changed. If not, it seems out of place in this figure.

Figure 6 components should be labeled a-d, and referenced in the text. The color scale is misleading. I suggest a divergent color scale with a neutral color at 0% change, so increases are clearly different from decreases.

The left two panels of figure 6 seem to fit better with the section describing the data. They do not depend on the estimated model like the right two panels.

Benzell, Seth G., Avinash Collis, and Christos Nicolaides. "Rationing social contact during the COVID-19 pandemic: Transmission risk and social benefits of US locations." Proceedings of the National Academy of Sciences (2020).

Chetty, Raj, John N. Friedman, Nathaniel Hendren, and Michael Stepner. "Real-time economics: A new platform to track the impacts of COVID-19 on people, businesses, and communities using private sector data." NBER Working Paper 27431 (2020).

Reviewer #2: This paper seeks to use mobile GPS data to analyze travel behavior changes for business clusters in Somerville MA. It also develops two scenarios to examine the future implications of another shutdown or a full reopening.

Main concerns:

1. The study objective is to examine the changes of travel behavior, as the title, abstract, introduction emphasize, however, this paper actually focuses on the change of the visit - the customer flows - to these business clusters.

The authors describe “… such as aggregated visit count as well as dwell time spent in various destinations” in their Data section, suggesting that the data from the SafeGraph only can provide the visit data, rather than the individual-level behavioral data, which largely confuses the readers about the research subjective of the study.

If focusing on behavioral changes, collecting the data for each resident is necessary, for example, if the dataset provides a unique GPS_User_ID, then using the ids to track their shopping and other business activities behaviors in 2019 and 2020, before and after restrictions, are appropriate to answer the research questions.

However, if using the aggregated flow data, how to distinguish random visitors to these destinations from the regular customers? The whole paper emphasizes many times about the OD, trip behavior choice and changes, impacts on future trip visits to these destinations, which strongly implies the study objective should be the individual level behavior changes and their influences on the aggregated level visit numbers to the same destinations.

Thus, a more appropriate use of the aggregated flow data to the business should be developing a model to predict the future visits considering the impacts of the COVID-19. However, using year 2019 data only will be a limiting point in this direction, since the prediction accuracy will be largely affected by the amount of historical data. Going this research direction will need to collect more historical data and put the 2020 shutdowns/restrictions as factors to the model to predict future flow changes.

If the authors want to predict individual behavior changes based on their current data, they can find the user_id to re-process their data and do the prediction. Currently, machine learning models or deep learning models can help to do a good job, for example, Zhu et al., 2017. Then, they can stick to their research questions about evaluating the systematic impacts on these business destination clusters based on the residents’ behavior changes. Of course, they can consider the random factors from tourism visitors to these destinations to make the future prediction model more accurate.

Reference:

Zhu, L., Gonder, J., & Lin, L. (2017). Prediction of individual social-demographic role based on travel behavior variability using long-term GPS data. Journal of Advanced Transportation, 2017.

2. The modeling section: nowadays, behavioral models are more suitable to analyze travel behavior changes, rather than discrete models. Since point 1 already describes some main concerns regarding the dataset, and the feasibility of using such data to answer certain questions. I recommend the authors take a deep examination of the GPS dataset (as well as the built environment data, policy change data, e.g., the widely use and provision of the masks in a lot of businesses, which might be helpful in the prediction), so that to have a better understanding of the data limits, characteristics, and research capacities.

3. Also, a detailed analysis of the demographic and economic characteristic of the GPS data population is needed, especially the study area is not a big area, such as multiple cities or the whole nation. If only simply describe - everyone nowadays has a phone so they are captured by the SafeGraph company - should not be acceptable. A more rigorous analysis of the data’s representativeness is very essential in this direction of research on localized analysis in particular.

6. PLOS authors have the option to publish the peer review history of their article (what does this mean?). If published, this will include your full peer review and any attached files.

Reviewer #1: No

Reviewer #2: No

---

## [Author Response · Author response to Decision Letter 0]

21 Apr 2021

Please see the attached Response to Reviewers.

---

## [Decision Letter · Decision Letter 1]

17 May 2021

PONE-D-20-35811R1

The impact of COVID-19 on trips to urban amenities: Examining travel behavior changes in Somerville, MA

PLOS ONE

Dear Dr. Sevtsuk,

Thank you for submitting your manuscript to PLOS ONE. After careful consideration, we feel that it has merit but does not fully meet PLOS ONE’s publication criteria as it currently stands. Therefore, we invite you to submit a revised version of the manuscript that addresses the points raised during the review process. The changes are minor and we hope that you will be able to address them soon.

We look forward to receiving your revised manuscript.

Kind regards,

Gabriel A. Picone

Academic Editor

PLOS ONE

Journal Requirements:

Reviewers' comments:

Reviewer's Responses to Questions

**Comments to the Author**

1. If the authors have adequately addressed your comments raised in a previous round of review and you feel that this manuscript is now acceptable for publication, you may indicate that here to bypass the “Comments to the Author” section, enter your conflict of interest statement in the “Confidential to Editor” section, and submit your "Accept" recommendation.

Reviewer #1: (No Response)

2. Is the manuscript technically sound, and do the data support the conclusions?

Reviewer #1: Yes

3. Has the statistical analysis been performed appropriately and rigorously? 

Reviewer #1: Yes

4. Have the authors made all data underlying the findings in their manuscript fully available?

Reviewer #1: Yes

5. Is the manuscript presented in an intelligible fashion and written in standard English?

Reviewer #1: Yes

6. Review Comments to the Author

Reviewer #1: The authors have significantly modified the manuscript and presentation of the results, which have improved the manuscript. I only have a few minor comments.

I like the new distance elasticities as a normalization to compare across time. However, the text does not describe how standard errors are calculated (e.g., delta method?, bootstrap?), which impacts inference.

The authors provide an improved description of the model. One of the key insights of the BLP model is that price is endogenous because of the unobservables in the error term. The authors use distance as a proxy for travel cost (price). You might add an explanation for why your travel cost version of the model is not subject to the same criticism. Or, if it still may be, how you would expect it to affect the estimates (if bias, which direction).

7. PLOS authors have the option to publish the peer review history of their article (what does this mean?). If published, this will include your full peer review and any attached files.

Reviewer #1: No

---

## [Author Response · Author response to Decision Letter 1]

19 May 2021

Please see the attached letter responding to Reviewer #1's two comments.

---

## [Editor Report · Decision Letter 2]

24 May 2021

The impact of COVID-19 on trips to urban amenities: Examining travel behavior changes in Somerville, MA

PONE-D-20-35811R2

Dear Dr. Sevtsuk,

We’re pleased to inform you that your manuscript has been judged scientifically suitable for publication and will be formally accepted for publication once it meets all outstanding technical requirements.

Kind regards,

Gabriel A. Picone

Academic Editor

PLOS ONE
---

## [Editor Report · Acceptance letter]

21 Jul 2021

PONE-D-20-35811R2 

The impact of COVID-19 on trips to urban amenities: Examining travel behavior changes in Somerville, MA 

Dear Dr. Sevtsuk:

I'm pleased to inform you that your manuscript has been deemed suitable for publication in PLOS ONE. Congratulations! Your manuscript is now with our production department. 

Kind regards, 

on behalf of

Dr. Gabriel A. Picone 

Academic Editor

PLOS ONE